# Modeling the future of HIV in Turkey: Cost-effectiveness analysis of improving testing and diagnosis

Emine Yaylali[1]*, Zikriye Melisa Erdogan[1], Fethi Calisir[1,2], Deniz Gokengin[3], Volkan Korten[4], Fehmi Tabak[5], Yesim Tasova[6], Serhat Unal[7], Berna Ozelgun[8], Tahsin Gokcem Ozcagli[8], Toros Sahin[8]

1 Faculty of Management, Department of Industrial Engineering, Istanbul Technical University, Istanbul, Turkey, 2 Nar Innovative Solutions, Istanbul, Turkey, 3 Faculty of Medicine, Department of Infectious Diseases and Clinical Microbiology, Ege University, Izmir, Turkey, 4 Faculty of Medicine, Department of Infectious Diseases, Marmara University, Istanbul, Turkey, 5 Faculty of Medicine, Department of Infectious Diseases, Istanbul University–Cerrahpasa, Istanbul, Turkey, 6 Faculty of Medicine, Department of Infectious Diseases, Cukurova University, Adana, Turkey, 7 Faculty of Medicine, Department of Infectious Diseases and Clinical Microbiology, Hacettepe University, Ankara, Turkey, 8 Gilead Sciences, Istanbul, Turkey

* emineyaylali@itu.edu.tr

## Abstract

### Aims

This study aimed to determine HIV incidence and prevalence in Turkey and to estimate the cost-effectiveness of improving testing and diagnosis in the next 20 years.

### Background

HIV incidence in Turkey has been rapidly increasing in the last decade with a particularly high rate of infection for younger populations, which underscores the urgent need for a robust prevention program and improved testing capacity for HIV.

### Methods

We developed a dynamic compartmental model of HIV transmission and progression among the Turkish population aged 15–64 and assessed the effect of improving testing and diagnosis. The model generated the number of new HIV cases by transmission risk and CD4 level, HIV diagnoses, HIV prevalence, continuum of care, the number of HIV-related deaths, and the expected number of infections prevented from 2020 to 2040. We also explored the cost impact of HIV and the cost-effectiveness of improving testing and diagnosis.

### Results

Under the base case scenario, the model estimated an HIV incidence of 13,462 cases in 2020, with 63% undiagnosed. The number of infections was estimated to increase by 27% by 2040, with HIV incidence in 2040 reaching 376,889 and HIV prevalence 2,414,965 cases. Improving testing and diagnosis to 50%, 70%, and 90%, would prevent 782,789,

**Data Availability Statement:** The data supporting the findings of this study are available within the

article or its supplementary materials. The data extracted from the three patient cohorts were anonymized, summarized and submitted in the data source file which does not contain any identifying information. The principal investigators of the cohorts, who are also co-authors of the submitted study permitted the use of cohort data for the study.

**Funding:** This project is supported by Gilead Sciences.

**Competing interests:** The authors have declared that no competing interests exist.

2,059,399, and 2,336,564 infections-32%, 85%, and 97% reduction in 20 years, respectively. Improved testing and diagnosis would reduce spending between $1.8 and $8.8 billion.

## Conclusions

In the case of no improvement in the current continuum of care, HIV incidence and prevalence will significantly increase over the next 20 years, placing a significant burden on the Turkish healthcare system. However, improving testing and diagnosis could substantially reduce the number of infections, ameliorating the public health and disease burden aspects.

## Introduction

The cumulative number of reported HIV infections in Turkey by the end of November 2020 was 27,767 with an annual number of 4,037 new diagnoses in 2020 [1]. Although the HIV prevalence in Turkey is still low (0.1–0.3%) based on the numbers reported by the Turkish Ministry of Health (MoH), the number of new diagnoses within the last five years has been rapidly increasing with a particularly high rate of infection for the younger population [2]. New HIV diagnoses increased threefold in the last decade, and the number of new diagnoses in the last five years accounted for 63% of the cumulative number reported so far (in S1 Fig in S1 File) [1]. If this trend continues, HIV prevalence in Turkey could significantly increase, overburdening the Turkish healthcare system.

In 2019, the Turkish MoH released three main goals in the national plan to respond to the HIV epidemic in Turkey in the context of UNAIDS' 90-90-90 goals [3]. The first goal was to reduce the number of new cases and HIV-related deaths by enhancing access to and the number of outreach and testing centers, as well as increasing the number of people who are diagnosed, initiated antiretroviral treatment (ART), and achieved virologic suppression. Although many studies reported hospital or region-based based estimates on HIV epidemiology in Turkey [4–7], national-level estimates of the incidence and prevalence of HIV in the Turkish population are lacking. Furthermore, although several studies reported 48–50% for the first 90, 86–88% for the second 90, and 85–87% for the last 90 of the continuum in Turkey, it is difficult to translate their results on a national level [8,9]. These data suggest a large gap for improvement, particularly for testing and diagnosis in Turkey, and the impact of such improvement has not been quantified in any study yet to the best of our knowledge.

Mathematical modeling is a common method to understand the spread of infectious diseases and to forecast the incidence for the short and long term in many infectious diseases, including HIV [10]. Modeling is an established and effective way of predicting epidemics and understanding the best ways of prevention. Models provide insights into the future of disease and it also allows policymakers to simulate possibilities under different what-if scenarios and assumptions. While randomized controlled trials are the gold standard in establishing the effectiveness of a prevention or treatment method, they are costly, time-consuming, and difficult to run. Due to budget constraints, the structure of the health system, and the long-term nature of our forecast, we preferred modeling over other techniques.

Sayan et al. developed a dynamic compartmental model of HIV to predict HIV/AIDS cases in Turkey between 1985 and 2016 and to estimate the basic reproduction number ($R_0$) [11], followed by a second study assessing the impact of awareness on the spread of disease [12]. In another modeling study, Mete et al. evaluated 90-90-90 targets in a patient cohort in Istanbul, the most populated city in Turkey, using the modeling tool of the European Center for Disease

Control and Prevention (ECDC) [13]. While both studies serve as examples of modeling studies that focus on HIV in Turkey, a more comprehensive model including risk populations, the continuum of care, cost, and economic implications would bring more perspective to the current situation.

The annual healthcare cost of people living with HIV (PLWH) in Turkey has been estimated to be around $4,000 to $5,000 in 2017 in a small cohort of 153 patients, varying based on patient characteristics [14]. In addition, the implications of keeping the status quo in HIV prevention, diagnosis, and treatment have not been assessed on disease burden and costs for Turkey. The aim of this study was to forecast the future of HIV in the next 20 years, to explore the impact of improving testing and diagnosis at different levels on both incidence and treatment costs, and to evaluate the cost-effectiveness of improved testing compared to the status quo.

## Materials and methods

### Model overview

We used a dynamic compartmental model of HIV transmission and progression in Turkey. The model considers a time horizon from 2005 to 2019 as its calibration period, and 2020 and beyond as its prediction period. It is validated against the number of confirmed cases by the MoH. The model simulates the Turkish population aged 15–64 years stratified by disease status (HIV negative and HIV positive) and transmission risk [men who have sex with men (MSM), people who inject drugs (PWID), and heterosexuals (HET)]. The HIV-positive population in the model is further divided into subpopulations based on disease stages (CD4≥200 and CD4<200) and continuum of care (undiagnosed, diagnosed but not on ART, on ART with no viral load suppression (VLS) and, on ART with VLS).

The model was formulated as a set of differential equations, and the equations were solved in MATLAB software (MathWorks; Natick, MA) using the ODE45 function without a predefined time step. The Supplemental Appendix in S1 File describes the structure and the formulations of the model, details of the data sets, model inputs, calibration methods, costs, and related calculations.

### HIV disease transmission, progression, and treatment

We modeled HIV transmission via sexual contact and needle sharing. The rate of transmission is reversely estimated by the reported number of HIV cases by the MoH due to a lack of data on sexual and needle-sharing behaviors such as the number of sexual/needle-sharing partners, condom use, and mixing in Turkey. Disease progression was based on the ART status and time spent in each disease stage [15]. We assumed ART initiation regardless of CD4 level based on the current guidelines and practice in Turkey and the proportion of diagnosed PLWH starts treatment was estimated from the patient cohorts. We assumed that patients who achieved VLS can improve their disease status, i.e., CD4 level could improve while on ART and VLS. We modeled treatment adherence by allowing patients to drop out of ART and VLS. The benefits of ART include better quality of life, reduced mortality, and disease transmission. We assumed a 53% of reduction in transmission if the infected person is aware of their disease status [16] and a 96% reduction if on ART or achieved VLS [17]. We included mortality due to HIV and non-HIV-related deaths.

### Data sources

We used data from the three major HIV patient cohorts in Turkey, including 8,566 HIV-positive patients (43% of all confirmed cases between 2005 and 2018 and 50% of all cases between

2011 and 2018). While the study period included January 1, 2011, to May 31, 2019, we also utilized earlier records collected in one data set from 2005 to 2011 to calibrate several parameters of the model for the first time period matching those dates. The multi-center and multi-regional cohorts included in the study are the largest HIV patient cohorts in Turkey, covering more than 30 cities. Overall, the median CD4 level at diagnosis was 373 cells/mm$^3$, and male patients accounted for 88% of the total population. The distribution of transmission groups was 46% heterosexuals, 31% MSM, 5% MSM and PWID, and 18% unidentified. Of the total population, 73% percent was on ART.

The data supporting the findings of this study are available within the article or its supplementary materials. The data extracted from the three patient cohorts were anonymized, summarized, and submitted in the data source file which does not contain any identifying information. The principal investigators of the cohorts, who are also co-authors of the submitted study permitted the use of cohort data for the study. Data extracted from cohort databases were anonymized before access and analysis, thus no informed consent and/or a consent waiver were obtained during the study. Ethics approval was not required for this modeling study.

## Model parameters

The model is populated with demographic, epidemiological, behavioral, and clinical data. Input parameters included population size by transmission risk group, the continuum of care, mortality rate (HIV-related and non-HIV related), HIV progression between disease stages, annual HIV cost per person by disease stage and continuum of care, and reduction in HIV transmission after diagnosis and treatment (Table 1). Parameters were estimated based on several sources such as the medical literature, the Turkish Statistical Institute, and the three major patient cohorts mentioned above. Input parameters that could not be estimated from data sets and literature or with limited data have been calibrated.

## Costs

HIV-related costs excluding ART costs were extracted from the literature. All costs were from a payer perspective, and all costs were converted to a 2020 baseline using the health component of the Turkish Consumer Price Index. Costs of ART were estimated from different drug regimens considering two price points: (i) regular pharmacy and (ii) governmental health insurance price. Drug prices were weighted with the percentage of patients in two patient cohorts on the relevant regimen. Since the majority of patients were in the governmental insurance system, we estimated the weighted average cost of treatment as 98% of governmental and 2% of regular pharmacy price. These percentages were obtained from the patient cohorts. The total direct HIV costs including the annual ART cost were $3,200, $3,217, and $5,218 for patients with CD4 levels >300 cells/mm$^3$, between 100 and 300 cells/mm$^3$, and <100 cells/mm$^3$, respectively (S3 Table in S1 File).

## Scenario analysis and model outcomes

To assess the effect of improving testing and diagnosis, we developed three scenarios where additional funding would improve the percentage of diagnosis to achieve the national goals by 2024 [3] as well as the UNAIDS 90-90-90 targets as follows: (i) low scenario: achieving 50% diagnosed in 2024 (ii) medium scenario: achieving 70% diagnosed in 2024 and (iii) high scenario: achieving 90% diagnosed in 2024. We examined the changes in the number of HIV infections, deaths, and costs associated with HIV from 2020 to 2040 under different scenarios. The base case scenario assumes the current continuum of care level is maintained after 2020.

**Table 1. Key model parameters.**

| Parameters | Value | Source |
|---|---|---|
| Population parameters | | |
| Size of the initial population by transmission group | | |
| HET | 46,863,864 | Calculated from Yıldırım, 2011 [18] and Data Set |
| PWID | 4,239 | Data Set |
| MSM | 69,897 | Data Set |
| Percentage of the initial population living with HIV by disease stage, % | | |
| CD4≥200 | 72.6 | Calculated from the number of diagnosed cases (Ministry of Health) [1] |
| CD4<200 | 27.4 | Calculated from the number of diagnosed cases (Ministry of Health) [1] |
| Distribution of the initial population living with HIV across the continuum of care, % | | |
| Diagnosed among CD4 ≥ 200 PLWH | 47.8 | Calibrated |
| Diagnosed among CD4 < 200 PLWH | 47.8 | Calibrated |
| on ART among CD4 ≥ 200 PLWDH | 70 | Data Set |
| on ART among CD4 < 200 PLWDH | 87 | Data Set |
| VLS among on ART PLWDH | 92 | Data Set |
| Entry rate (Birth rate) | 1.82 | Turkish Statistical Institute (TÜİK) [19] |
| The non-HIV-related mortality rate | 0.59 | Turkish Statistical Institute (TÜİK) [19] |
| HIV-related mortality rate, % | | |
| The HIV-related mortality rate among undiagnosed | 1.02 | Assumed the same with a diagnosis, not on ART |
| The HIV-related mortality rate among those diagnosed not on ART | 1.02 | Data Set |
| The HIV-related mortality rate among those diagnosed on ART, not VLS | 0.93 | Data Set |
| The HIV-related mortality rate among VLS | 0.83 | Data Set |
| HIV Progression (rates across horizontal compartments) | | |
| The natural length of time by disease stage if not on ART | 32 years | CDC Vital Signs [15] |
| Rate of disease stage improvement while undiagnosed & diagnosed not on ART | 8.54% | Calibrated |
| Length of time if on ART | 71 years | CDC Vital Signs [15] |
| Rate of disease stage improvement if on ART not VLS | 1.39% | Calculated from CDC Vital Signs [15] |
| Length of time if on ART and VLS | 80 years | [20] |
| Rate of disease stage improvement while VLS | 20.7% | Calibrated |
| Continuum of care progression (rates across vertical compartments), % | | |
| Percentage of PLWH that are diagnosed | 16.47 | Calibrated |
| Percentage of diagnosed PLWH that are on ART | 70%- 87% | Data Set |
| Percentage of diagnosed and on ART PLWH that achieve VLS | 85.41%-94.39% | Data Set |
| Percentage of dropping out of VLS | 2.31% | Data Set |
| Infectivity (Force of infection) | Calibrated | |
| HIV prevalence rate by transmission group | | |
| HET | 0.01% | Data Set |

(*Continued*)

**Table 1.** (Continued)

| Parameters | Value | Source |
| --- | --- | --- |
| PWID | 0.66% | [21] |
| MSM | 3% | [22] |
| Reduction in HIV transmission if on ART | 96% | [17] |
| Reduction in HIV transmission if on VLS | 96% | [17] |
| Reduction in HIV transmission if the infected person is aware of disease status vs unaware | 53% | [23] |

HET: Heterosexuals.

PWID: People who inject drugs.

MSM: Gay, bisexual, and other men who have sex with men.

PLWH: People Living with HIV.

PLWDH: People living with diagnosed HIV.

ART: Antiretroviral therapy.

VLS: Viral load suppression.

The model generated the number of new HIV cases by transmission risk and CD4 level, HIV diagnoses, HIV prevalence, continuum of care, the number of HIV-related deaths, and the expected number of infections prevented. We also collected cost-related outcomes such as total direct and indirect HIV costs, cost per infection prevented, incremental cost-effectiveness ratios (ICER) for diagnosis scenarios, and return on investment (ROI).

## Cost-effectiveness analysis

Due to limited data, the model was validated against the number of diagnosed cases and HIV-related deaths reported by the MoH for 2019. In addition, our continuum of care results was compared with previously published estimates. The predicted number of diagnosed cases and HIV-related deaths were within 5% of target real data values. While the undiagnosed percentage was consistent with other reported studies, the percentage of PLWH on ART was lower in our study, therefore underestimating the benefits of improving testing and diagnosis.

For all scenarios, we estimated the number of HIV cases, total ART, and care costs and determined the ICER of improving percent diagnosed in comparison with the base case scenario. A twenty-year time horizon was used, and the cumulative direct and indirect costs of all scenarios were forecasted. Future costs and effects are discounted at a 3% discount rate as suggested by the Panel on Cost-Effectiveness in Health and Medicine [24].

## Sensitivity analysis

We conducted a one-way sensitivity analysis to quantify the effect of model parameters such as the force of infection, the continuum of care, and mortality rate on the total 20-year HIV incidence. We varied all model parameters, including the calibrated inputs by ±20% from their base values, and collected the parameters with the largest effect on HIV incidence. We also performed the elementary effects method to observe the non-linear effects and interactions between model parameters. Further details of the elementary effects method are presented in S1 File.

## Results

Under the base case scenario, the model estimated the HIV incidence of around 13,462 cases in 2020, with 63% being undiagnosed. HIV incidence was estimated to reach 72,844 in 2030 and

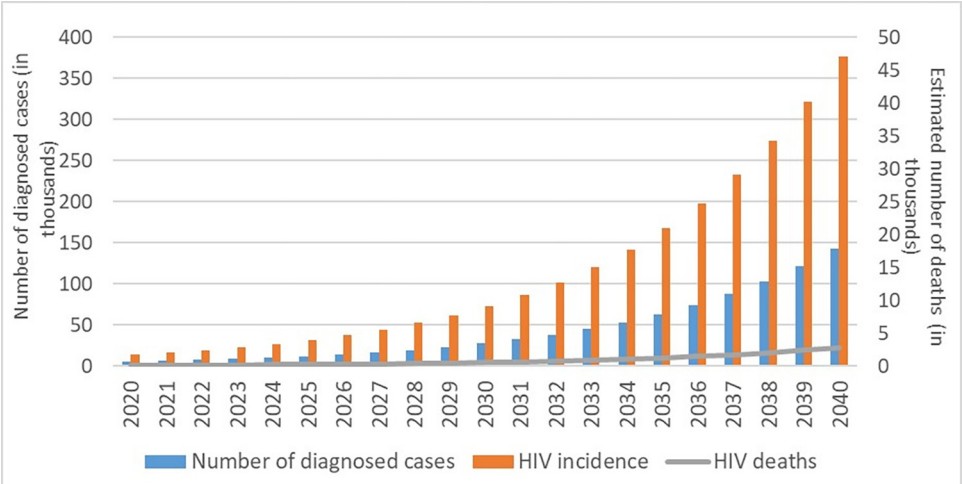

**Fig 1. Estimated number of HIV incidence, diagnosis, and HIV-related deaths, 2020–2040.**

surpass 376,889 infections in 2040 with a 27% increase (Fig 1). Model outcomes suggested that HIV prevalence would be around 2,414,965 cases by 2040 in Turkey if the current trend continued.

For the base case scenario, around 44% of PLWH would be diagnosed in 2020, and a 2% increase would be expected by 2040 for those with CD4 $\geq$200, while it would remain the same for those with CD4 <200. The percentages of diagnosed persons using ART who achieved VLS were predicted to be 77.5% (82% for CD4$\geq$200 and 73% for CD4 <200) and 83% in 2020, respectively.

In the case of improving testing and diagnosis rates to 50%, 70%, and 90%, infections prevented will increase to 782,789 (32% reduction), 2,059,399 (85% reduction), and 2,336,564 (97% reduction) in 20 years, respectively (Fig 2).

## Cost-effectiveness analysis

Since the number of infections would be reduced with an improved diagnosis rate, total ART and care costs also would be reduced by between $1.8B and $4.7B as compared with the next best case in the scenarios. Cost per infection averted was around $3,696, $5,728, and $12,269 for low, medium, and high scenarios, respectively (Table 2). Although all three scenarios were cost-saving, the additional cost of improving diagnosis was not included in the model, which might have had implications on the results. Although considering the cost-effectiveness threshold could be estimated at around $8,597 in 2020 based on Turkey's per-capita gross domestic product (GDP), there is a significant cost that could be incurred before reaching ICERs that is greater than the threshold.

ROI was used to compare the efficiency of these strategies. Although the three scenarios did not display significant differences in terms of ROI, the largest ROI was in the high scenario in which ART and care costs were offset due to the reduction of HIV incidence by 2025 (Fig 3). Other scenarios reached positive ROI one or two years later. The initial investment would have been doubled when the ROI rate was above 100%. In our study, the ROI rate of the high scenario would exceed this threshold by 2029, which was the earliest among all scenarios.

## Sensitivity analysis

One-way sensitivity analysis allowed us to rank the parameters by their importance and identify the top 10 parameters that the model was most sensitive to (Fig 4). The force of infection,

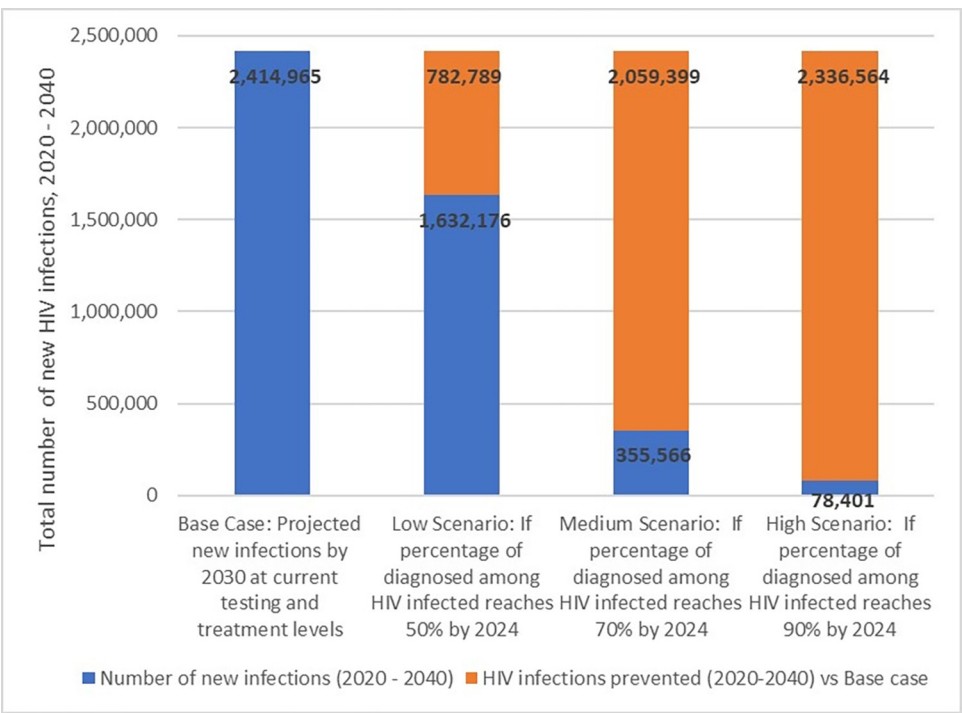

**Fig 2. The total number of new infections and HIV infections prevented for low, medium, and high scenarios, 2020–2040.**

which is the rate of infection per uninfected person, was found to be the most important parameter of all, while the annual diagnosis rate and the percentage of diagnosed PLWH played a crucial role in the model as well. They were followed by the reduction in HIV transmission, annual treatment rates, the percentage of CD4≥200 among PLWH, and the non-HIV-related mortality rate. The less significant parameters are presented in S4 Fig in S1 File.

According to the elementary effects method that determined mean absolute and standard deviations of elementary effects, parameters that involved nonlinear and/or interaction effects

**Table 2. Cost-effectiveness of improving the percentage of diagnosed among HIV positive persons to 50%, 70%, and 90% by 2024.**

|  | Discounted number of new infections (2020–2040) | Discounted HIV infections prevented vs next best | Total ART and care cost (2020–2040), $ | Discounted total ART and care cost prevented, $[a] | Cost per infection averted, $ | ICER, $ vs next best |
|---|---|---|---|---|---|---|
| **Base Case** | 1,560,067 |  | 12,974,296,921 |  |  | Cost-saving |
| **Low Scenario** | 1,073,913 | 486,154 | 11,177,545,651 | 1,796,751,270 | 3,696 | Cost-saving |
| **Medium Scenario** | 259,785 | 814,128 | 6,513,861,565 | 4,663,684,086 | 5,728 | Cost-saving |
| **High Scenario** | 65,597 | 194,188 | 4,131,385,091 | 2,382,476,474 | 12,269 | Cost-saving |

ART: Antiretroviral therapy.

ICER: Incremental cost-effectiveness ratio.

[a] Cost prevented is calculated for the next best strategy.

ICER = (Cost$_A$- Cost$_B$) / The number of infections prevented (B-A).

Discounted using 3%.

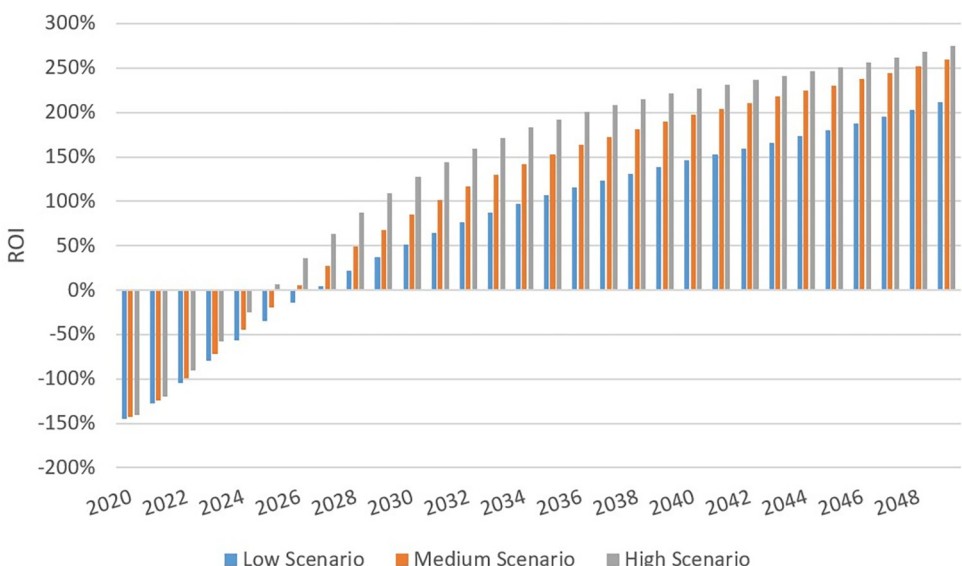

**Fig 3. Cumulative return on investment (ROI).** Although all scenarios showed positive ROI, the high scenario had the best performance.

due to high mean and standard deviations were as follows: the force of infection for HET, reduction in HIV transmission for diagnosed PLWH and for PLWH on ART, rate of disease stage improvement for the compartment of undiagnosed PLWH with CD4$\geq$200 cells/mm$^3$, and non-HIV related death rates while other parameters had a negligible effect on the model (Fig 5).

## Discussion

In this study, we developed a mathematical model that simulates HIV transmission and progression in Turkey and analyzed the impact of the continuum of care on the future of the

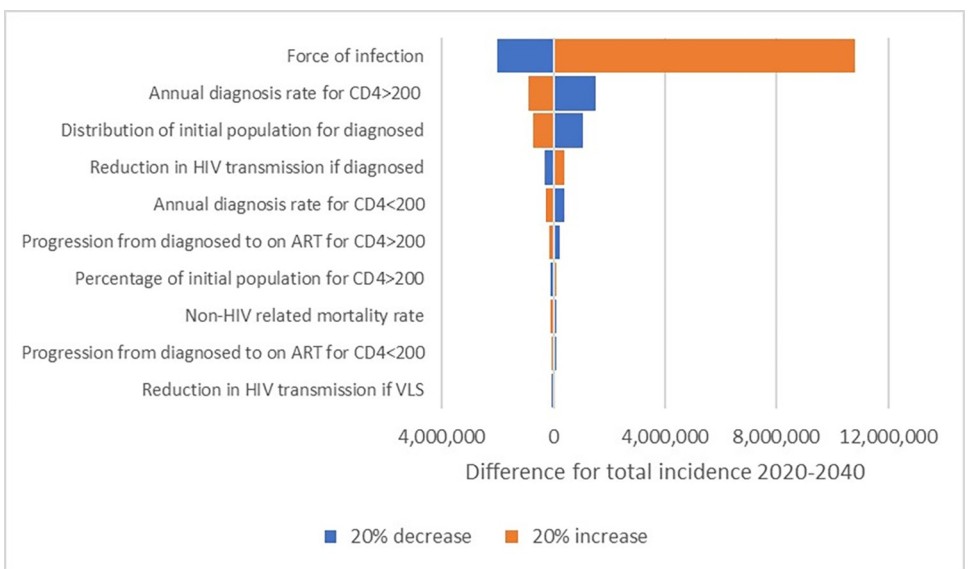

**Fig 4. Tornado diagram showing the top 10 parameters.** Abbreviations: VLS: Viral load suppression, ART: Antiretroviral treatment.

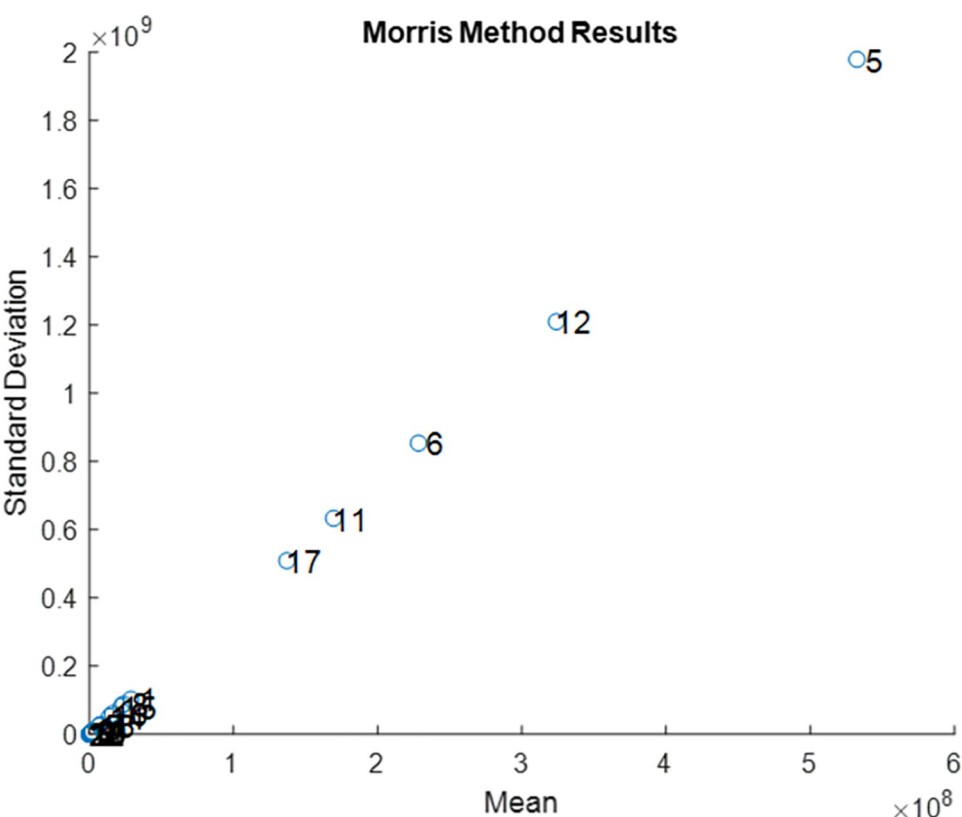

**Fig 5. Elementary effects analysis results of model parameters based on the total HIV incidence between 2020 to 2040.**

epidemic. Our results showed that HIV incidence and prevalence might significantly increase in the next 20 years resulting in a substantial burden on the healthcare system if there is no improvement in HIV awareness levels, access to diagnosis, and the continuum of care. Based on a previous similar mathematical modeling study with three compartments, the estimated number of patients was increasing rapidly and exceeded the real-life cases after 1994 [11]. The continuum of care estimations in our study was similar to those from other published studies ranging between 48–50% for the first 90, 75.3–88% for the second 90, and 85–87% for the last 90 of the continuum [8,9]. Although a study including PLWH in Istanbul reported much higher percentages for the first two elements of the continuum (73% and 92%, respectively) [13], the results are not comparable due to the unique characteristics of the Istanbul population, which is not representative of the overall HIV population in Turkey.

The results of this study suggest that the largest gap in the continuum in Turkey is the diagnosis element, which underscores the urgent need for rapid enhancement of access to HIV testing and removing the barriers to testing. Improving HIV diagnosis was shown to have a significant impact on the future of HIV in Turkey. Achieving 50% diagnosed among PLWH will prevent 32% of new infections, and reaching the UNAIDS target of 90% diagnosed will nearly eliminate HIV in Turkey by decreasing HIV prevalence to less than 80,000 cases over 20 years. However, our model estimated a much lower rate of diagnosis in Turkey, whereas globally, the HIV diagnosis rate is 81% [25]. There seems to be a major gap between reality and intended goals. As the spread of HIV continues, so makes its impact on social capital, population structure, and economic growth.

The current strategy of testing in the country is testing of the general population through testing for blood and organ donations, premarital testing, and testing prior to major surgery. Key populations are not well defined, characterized, and included in response to the epidemic, and innovative testing strategies for specific populations are missing. However, previous studies suggest high prevalence rates among vulnerable populations and specifically MSM, and transmission through male-to-male sex has become more dominant in the last decade [1,2,7,26]. In addition, there is still a lot of stigmas around testing, and it is clear that Turkey needs innovative strategies to address barriers to testing and to improve its testing capacity.

HIV management is very costly, and the annual direct costs of a diagnosed HIV patient in this study were estimated to vary between $1,280 and $4,196 (S4 Table in S1 File) based on CD4 level and continuum of care, which was comparable to the results of a previous study [27]. The majority of the cost was attributed to ART cost, while as expected, costs for patients with CD4$\geq$200 cells/mm$^3$ were less than those for patients with CD4 <200 cells/mm$^3$ in both studies [27]. Turkey is known to pay the lowest price to original antiretrovirals throughout Europe, and the price of generic drugs is evened out with the originals after an initial discount in the original price following the launch of the generic, which does not allow a further benefit. Although the annual cost per person is significantly lower than the cost in the US [28] and Europe [29], the cumulative burden of the total cost with the predicted numbers in the following years is likely to be devastating for the fragile economy of the country.

Our results suggest that improving the diagnosis rate would be cost-saving. Although enhancing diagnosis would require additional costs such as new testing centers, development of outreach programs, and awareness campaigns, the intervention would still be cost-effective if kept under the threshold of $8,597. For each HIV case prevented, around $3,696 would be saved from treatment and care costs when the rate of diagnosis is increased by only 6% by 2024; this is an achievable goal. Marginal change in cost per infection prevented increases significantly to $5,728 and $12,269 with the medium and high scenarios, which require a lot more work and effort.

New HIV infections are on the rise for younger ages in Turkey with the main transmission routes including mostly heterosexual intercourse (33%) followed by homosexual intercourse (15%) [1,7]. There is also a significant percentage of PLWH with unknown or missing transmission routes (50%), many of which can be attributed to MSM. A recent study including Central European countries showed that the most prominent increase in transmission modes was among MSM with the highest number reported from Turkey with a greater than the 10-fold increase between 2005 and 2014 [2]. Another study, including the largest cohort in Turkey, reported a significant recent rise in transmission by male-to-male sex [30]. Therefore, most at-risk populations are an important element of the HIV epidemic, and transmission through male-to-male sex is rapidly increasing in Turkey, which necessitates the inclusion of key populations in modeling studies to better project the future of HIV. To our knowledge, this is the first HIV modeling study that considers risk populations and evaluates the cost-effectiveness of diagnosis in Turkey.

This work can contribute to various aspects of the HIV epidemic. On the public health side, this is the first study that developed the HIV continuum of care on a national level. Furthermore, it is the first comprehensive mathematical model of HIV spread in Turkey that considers multi-faceted dynamics of transmission, including the likely key populations and prevention as well as the economic impact. The study facilitates the understanding of the relationships between testing/diagnosis rates and the number of infections by using scenario analysis and cost-effectiveness calculations. Therefore, it serves as a tool for public health decision-makers to see the big picture of the epidemic with projections for the future by comparing the 'status quo' versus 'low/medium/high scenarios.'

From the methodological view, this study complements others by including the key populations in the dynamic model structure, dealing with the economic impact by cost-effectiveness analysis, and analyzing the significance of the parameters with two different types of sensitivity analyses. Further research is necessary to determine the most cost-effective interventions to reduce the rapid increase in the number of new HIV cases in Turkey.

Our study has several limitations:

1. Data was limited or inadequate for several model parameters such as the force of infection and diagnosis rate. The model inputs have a direct impact on the number of infections estimated by the model. Hence, the quality of input parameters or the lack of data for estimation could yield uncertainty in results. To overcome this problem, the parameters were calibrated and a sensitivity analysis was conducted.

2. The model assumed that current behaviors would continue in the following 20 years. However, people tend to change their risk behavior patterns over the years; hence, the long-term projections of our model should be interpreted accordingly. Although we do not need any special assumptions for implementing the elementary effects method, we used our parameters as uniformly distributed among their sample space. Appropriate guidance is missing for the best value of the number of iterations and increase/decrease rate [31,32]. While the elementary effects method shows the non-linear and/or interaction effects of parameters in an efficient way, it does not present any measure for ranking parameters by their importance.

Dynamic compartmental disease models are the most widely used methods in mathematical epidemiology. While we utilized this type of model in our study, it could be further enhanced by adding uncertainty and adopted for stochastic epidemic modeling. To understand the effect of individual behaviors, agent-based simulation models can be employed in future studies. Furthermore, the best ways to achieve 50%, 70%, and 90% diagnosed rates in 2024 should be further studied and identified in order to design and implement effective strategies.

## Conclusions

Our study identifies some important opportunities for decision-makers. Improving the HIV diagnosis rate will be cost-saving in Turkey. If there is no improvement in the current continuum of care, HIV incidence and prevalence will significantly increase over the next 20 years and will place a significant burden on the Turkish healthcare system. However, improving strategies around prevention, testing, and diagnosis would substantially reduce the number of infections.

## Supporting information

**S1 File. Supplemental appendix.** More detailed information, such as model diagram and explanation, calibration process and cost calculations, was summarized in S1 File.
(DOCX)

**S2 File.**
(DOCX)

**S1 Data.**
(ZIP)

## Author Contributions

**Conceptualization:** Emine Yaylali, Toros Sahin.

**Data curation:** Emine Yaylali, Zikriye Melisa Erdogan, Deniz Gokengin, Volkan Korten, Fehmi Tabak, Yesim Tasova.

**Formal analysis:** Emine Yaylali, Zikriye Melisa Erdogan, Serhat Unal.

**Funding acquisition:** Fethi Calisir, Toros Sahin.

**Methodology:** Emine Yaylali, Zikriye Melisa Erdogan, Toros Sahin.

**Project administration:** Fethi Calisir, Toros Sahin.

**Software:** Emine Yaylali, Zikriye Melisa Erdogan.

**Supervision:** Berna Ozelgun, Tahsin Gokcem Ozcagli, Toros Sahin.

**Validation:** Deniz Gokengin, Volkan Korten, Fehmi Tabak, Yesim Tasova, Serhat Unal.

**Writing – original draft:** Emine Yaylali.

**Writing – review & editing:** Fethi Calisir, Deniz Gokengin, Volkan Korten, Fehmi Tabak, Yesim Tasova, Serhat Unal, Berna Ozelgun, Tahsin Gokcem Ozcagli, Toros Sahin.

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
