## [Decision Letter · Decision Letter 0]

24 Feb 2023

PONE-D-22-32071Modeling the future of HIV in Turkey: Cost-effectiveness analysis of improving testing and diagnosisPLOS ONE

Dear Dr. Yaylali,

Thank you for submitting your manuscript to PLOS ONE. After careful consideration, we feel that it has merit but does not fully meet PLOS ONE’s publication criteria as it currently stands. Therefore, we invite you to submit a revised version of the manuscript that addresses the points raised during the review process.

We look forward to receiving your revised manuscript.

Kind regards,

Hamufare Dumisani Dumisani Mugauri, Ph.D. Public Health

Academic Editor

PLOS ONE

Journal Requirements:

2. In the ethics statement in the manuscript and in the online submission form, please provide additional information about the patient records/samples used in your retrospective study. Specifically, please ensure that you have discussed whether all data/samples were fully anonymized before you accessed them and/or whether the IRB or ethics committee waived the requirement for informed consent. If patients provided informed written consent to have data/samples from their medical records used in research, please include this information.

3. In the Methods section of your manuscript, please confirm that all data sources you used were publicly available and anonymized. If this is not the case, please provide information on what permissions you were granted to access these data.

This project is supported by Gilead Sciences.

Additional Editor Comments:

A major revision of the manuscript is needed as a consequence of the reviewer comments.

Reviewers' comments:

Reviewer's Responses to Questions

**Comments to the Author**

1. Is the manuscript technically sound, and do the data support the conclusions?

Reviewer #1: Yes

Reviewer #2: Yes

Reviewer #3: Yes

2. Has the statistical analysis been performed appropriately and rigorously? 

Reviewer #1: Yes

Reviewer #2: Yes

Reviewer #3: Yes

3. Have the authors made all data underlying the findings in their manuscript fully available?

Reviewer #1: Yes

Reviewer #2: Yes

Reviewer #3: Yes

4. Is the manuscript presented in an intelligible fashion and written in standard English?

Reviewer #1: Yes

Reviewer #2: Yes

Reviewer #3: Yes

5. Review Comments to the Author

Reviewer #1: Summary comments to authors

The effectiveness was established through mathematical modelling of the different scenarios, including the status quo. No randomized controlled trial was done. This therefore may not present strong evidence of effectiveness. A randomized controlled trial would provide a gold standard. In addition, the authors would do well to provide justification of relying on mathematical modelling. The range of costs were presented in detail for all scenarios including the status quo. The costs were also wide enough for the research question. However, it would also have been helpful to have cost from the provider perspective, as the focus for these costs were from the payer’s perspective.

This would have provided enough evidence for real-world decision making. The paper covered the payer’s viewpoint, but did not cover the other relevant viewpoints, such as the community and providers. If these viewpoints were covered, then they should be clear and obvious to the readers. This information is important so as to get as much information as possible to make a strong inference. This may have had possible effects on the ICER. Capital costs and operating costs such as salaries for staff were not included. The study took into account costs of consumables such as test kits for CD4 and viral load and the antiretroviral medicines. It is important that the authors also revisit this particular aspect of the analysis, which may have an effect on the final result.

Reviewer #2: The article is well-written. The quantitative analysis of the paper, which is mainly based on a dynamic compartmental model, is well done. However, a part completely misses: indeed, the considered model should be described in more detail from a mathematical point of view, especially from the point of view of its formulation. Just as an example, details about the differential equations behind the model are not provided. Without this preliminary part, it is very difficult, for the reader, the comprehension of the estimates/results obtained by the authors.

Reviewer #3: The author did not factor in discounting into the model. This is a limitation as findings could vary substantially if a discounting rate was factored the costs and effects of the three scenarios modelled could be much lower than those presented.

6. PLOS authors have the option to publish the peer review history of their article (what does this mean?). If published, this will include your full peer review and any attached files.

Reviewer #1: **Yes: **Howard Nyika

Reviewer #2: No

Reviewer #3: **Yes: **Gerald Shambira

---

## [Author Response · Author response to Decision Letter 0]

26 Apr 2023

Dear Dr. Hamufare Dumisani Dumisani Mugauri,

Thank you for the opportunity to revise our manuscript. We appreciate the constructive comments provided by the reviewers. We have updated the paper to address all the concerns raised by you and the reviewers. In addition, we also updated the costs to 2020 values and presented the discounted results accordingly. The revisions made helped us improve our manuscript. Please see our detailed responses to the comments made by the reviewers. All edits to the manuscript are highlighted in blue font.

Reviewer #1, Comment 1:

The effectiveness was established through mathematical modelling of the different scenarios, including the status quo. No randomized controlled trial was done. This therefore may not present strong evidence of effectiveness. A randomized controlled trial would provide a gold standard. In addition, the authors would do well to provide justification of relying on mathematical modelling. 

Response to Reviewer #1, Comment 1:

Thank you for this comment. Indeed, randomized controlled trials are the gold standard for establishing the effectiveness of any prevention method. However, they are also costly, time-consuming, and difficult to establish. In Turkey, the majority of the health care system is managed by the government and a significant portion of HIV testing is provided by state hospitals. To conduct a randomized controlled trial on improving HIV testing and diagnosis would require state-level policymaking and a budget that would compete with other public health programs such as vaccinations. The main aim of this study was to inform the government of the rising HIV threat and to provide evidence on what the consequences of ignoring the problem would be in the long run. We hope that the results of our study would encourage policymakers to take action, including designing randomized controlled trials to determine the most cost-effective ways of reducing HIV incidence in Turkey. Another benefit of modelling over randomized trials is that it provides long-term forecasts. In our study, we project HIV incidence over a 20-year time period which would not be possible with a randomized controlled trial. 

To address this comment, we have added the following statement to the Introduction. 

“Modeling is an established and effective way of predicting epidemics and understanding the best ways of prevention. Models provide insights into the future of disease and it also allows policymakers to simulate possibilities under different what-if scenarios and assumptions. While randomized controlled trials are the gold standard in establishing the effectiveness of a prevention or treatment method, they are costly, time-consuming, and difficult to run. Due to budget constraints, the structure of the health system, and the long-term nature of our forecast, we preferred modeling over other techniques” 

Reviewer #1, Comment 2:

The range of costs were presented in detail for all scenarios including the status quo. The costs were also wide enough for the research question. However, it would also have been helpful to have cost from the provider perspective, as the focus for these costs were from the payer’s perspective.

This would have provided enough evidence for real-world decision making. The paper covered the payer's viewpoint, but did not cover the other relevant viewpoints, such as the community and providers. If these viewpoints were covered, then they should be clear and obvious to the readers. This information is important so as to get as much information as possible to make a strong inference. This may have had possible effects on the ICER. Capital costs and operating costs such as salaries for staff were not included. The study took into account costs of consumables such as test kits for CD4 and viral load and the antiretroviral medicines. It is important that the authors also revisit this particular aspect of the analysis, which may have an effect on the final result.

Response to Reviewer #1, Comment 2:

Thank you for this comment. We extensively searched the literature on HIV-related costs in Turkey during our study. Unfortunately, there were only two relevant studies one of which was a conference poster. We collected drug costs from the manufacturers and calculated ART costs based on the frequency of treatment regimens from the patient cohort data. Other costs such as test and lab costs and costs related to comorbidities etc. were from the study by Kockaya (2016) and they were adjusted to 2020 values [1]. We could not find any data on capital costs and operating costs since the majority of the hospitals in Turkey are state hospitals and the operating cost of these hospitals is not publicly available. We acknowledge that the provider or societal perspective would be more comprehensive than the payer perspective and it could have possible effects on the ICERs, however in the absence of data, we had to use the most inclusive cost aspects. 

To provide a larger perspective on cost, i.e. a truncated societal perspective, we added the results with the inclusion of indirect costs from a study by Tatar (2016) in the revised manuscript[2]. We calculated ICERs where the costs included productivity loss cost due to HIV and presented these results in Table S7 in the Supplemental Appendix (S1 File). 

Reviewer #2, Comment 1:

Reviewer #2: The article is well-written. The quantitative analysis of the paper, which is mainly based on a dynamic compartmental model, is well done. However, a part completely misses: indeed, the considered model should be described in more detail from a mathematical point of view, especially from the point of view of its formulation. Just as an example, details about the differential equations behind the model are not provided. Without this preliminary part, it is very difficult, for the reader, the comprehension of the estimates/results obtained by the authors.

Response to Reviewer #2, Comment 1:

Thank you for this comment. Details related to the differential equations behind the model are added to the Supplemental Appendix (S1 File) and Fig S2 is updated to present the relationship between the model parameters and compartments. 

Reviewer #3, Comment 1:

Reviewer #3: The author did not factor in discounting into the model. This is a limitation as findings could vary substantially if a discounting rate was factored the costs and effects of the three scenarios modelled could be much lower than those presented.

Response to Reviewer #3, Comment 1:

Thank you for this comment. We adjusted all costs and effects to 2020 dollars and discounted any future costs and effects at a 3% discount rate as suggested by the Panel on Cost-Effectiveness in Health and Medicine [3]. Table 2 and Figure 3 are updated based on the discounted results. In addition, we conducted a sensitivity analysis on the discounting factor with no discounting and a 5% discounting factor. Results of the sensitivity analysis are presented in Tables S5 and S6 in the Supplemental Appendix (S1 File). 

Additional Requirements:

Thank you. We ensured that the manuscript meets the style requirements including file naming.

2. In the ethics statement in the manuscript and in the online submission form, please provide additional information about the patient records/samples used in your retrospective study. Specifically, please ensure that you have discussed whether all data/samples were fully anonymized before you accessed them and/or whether the IRB or ethics committee waived the requirement for informed consent. If patients provided informed written consent to have data/samples from their medical records used in research, please include this information.

Thank you. We added the following statement to the Methods section and the cover letter.

“Data extracted from cohort databases were anonymized before access and analysis, thus no informed consent and/or a consent waiver were obtained during the study. Ethics approval was not required for this modeling study.”

3. In the Methods section of your manuscript, please confirm that all data sources you used were publicly available and anonymized. If this is not the case, please provide information on what permissions you were granted to access these data.

Thank you. We added the following statement to the Methods section and the cover letter.

"The data supporting the findings of this study are available within the article or its supplementary materials. The data extracted from the three patient cohorts were anonymized, summarized, and submitted in the data source file which does not contain any identifying information. The principal investigators of the cohorts, who are also co-authors of the submitted study permitted the use of cohort data for the study."

This project is supported by Gilead Sciences. 

Thank you. We added the following statement to the cover letter.

“This project is supported by Gilead Sciences. The funders had no role in study design, data collection and analysis, the decision to publish, or preparation of the manuscript.”

Thank you. We updated the manuscript and Supplemental Appendix (S1 File) according to the guideline. 

References

1. Kockaya G, Zengin TE, Yenilmez FB, Dalgic C, Malhan S, Cerci P, et al. Analysis of the treatment cost of HIV/AIDS in Turkey. Farmeconomia Heal Econ Ther pathways. 2016;17(1):13–7. 

2. Tatar M, Kockaya G, Ozelgun B, Zengin Elbir T, Senturk A, Tuna E, et al. Indirect Cost Of Hiv/Aids: Results Of A Survey From A Turkish Research Center. Value Heal. 2016;19(7):A411. 

3. Sanders GD, Neumann PJ, Basu A, Brock DW, Feeny D, Krahn M, et al. Recommendations for Conduct, Methodological Practices, and Reporting of Cost-effectiveness Analyses: Second Panel on Cost-Effectiveness in Health and Medicine. JAMA - J Am Med Assoc. 2016;316(10):1093–103.

---

## [Editor Report · Decision Letter 1]

12 May 2023

Modeling the future of HIV in Turkey: Cost-effectiveness analysis of improving testing and diagnosis

PONE-D-22-32071R1

Dear Dr. Yaylali,

We’re pleased to inform you that your manuscript has been judged scientifically suitable for publication and will be formally accepted for publication once it meets all outstanding technical requirements.

Kind regards,

Hamufare Dumisani Dumisani Mugauri, Ph.D. Public Health

Academic Editor

PLOS ONE

---

## [Editor Report · Acceptance letter]

23 Jun 2023

PONE-D-22-32071R1 

Modeling the future of HIV in Turkey: Cost-effectiveness analysis of improving testing and diagnosis 

Dear Dr. Yaylali:

I'm pleased to inform you that your manuscript has been deemed suitable for publication in PLOS ONE. Congratulations! Your manuscript is now with our production department. 

Kind regards, 

on behalf of

Mr Hamufare Dumisani Dumisani Mugauri 

Academic Editor

PLOS ONE